# HAUSP Is a Key Epigenetic Regulator of the Chromatin Effector Proteins

**DOI:** 10.3390/genes13010042

**Published:** 2021-12-24

**Authors:** Omeima Abdullah, Mahmoud Alhosin

**Affiliations:** 1College of Pharmacy, Umm Al-Qura University, Makkah 21955, Saudi Arabia; oaabdullah@uqu.edu.sa; 2Biochemistry Department, Faculty of Science, King Abdulaziz University, Jeddah 21589, Saudi Arabia; 3Centre for Artificial Intelligence in Precision Medicines, King Abdulaziz University, Jeddah 21589, Saudi Arabia

**Keywords:** HAUSP, UHRF1, epigenetic, cancer, tumor suppressor gene

## Abstract

HAUSP (herpes virus-associated ubiquitin-specific protease), also known as Ubiquitin Specific Protease 7, plays critical roles in cellular processes, such as chromatin biology and epigenetics, through the regulation of different signaling pathways. HAUSP is a main partner of the “Epigenetic Code Replication Machinery,” ECREM, a large protein complex that includes several epigenetic players, such as the ubiquitin-like containing plant homeodomain (PHD) and an interesting new gene (RING), finger domains 1 (UHRF1), as well as DNA methyltransferase 1 (DNMT1), histone deacetylase 1 (HDAC1), histone methyltransferase G9a, and histone acetyltransferase TIP60. Due to its deubiquitinase activity and its ability to team up through direct interactions with several epigenetic regulators, mainly UHRF1, DNMT1, TIP60, the histone lysine methyltransferase EZH2, and the lysine-specific histone demethylase LSD1, HAUSP positions itself at the top of the regulatory hierarchies involved in epigenetic silencing of tumor suppressor genes in cancer. This review highlights the increasing role of HAUSP as an epigenetic master regulator that governs a set of epigenetic players involved in both the maintenance of DNA methylation and histone post-translational modifications.

## 1. Introduction

Ubiquitination, the addition of a small protein called ubiquitin (76 amino acids) to other target proteins is a post-translational protein modification that controls almost all processes in the cell and leads to different outcomes, ranging from proteasomal protein degradation and cellular trafficking to cell proliferation, apoptosis, autophagy, DNA repair, and epigenetic modulation of gene expression [1,2,3,4]. Ubiquitination results from a successful collaboration between multienzyme cascades that involve E1 activating enzymes, E2 conjugating enzymes, and E3 ubiquitin ligases [1,5,6,7].

Deubiquitinases, or deubiquitinating enzymes (DUBs), are enzymes that protect many proteins from ubiquitination and are active in various pathologies, including cancer [8,9,10,11]. One of the well-documented cancer-associated DUBs is HAUSP (herpes virus-associated ubiquitin-specific protease), also known as Ubiquitin Specific Protease 7 (USP7), an enzyme that is overexpressed in many solid and blood malignancies [12,13,14,15,16]. The structure of HAUSP includes seven domains: the N-terminal TRAF-like (Tumor necrosis factor Receptor–Associated Factor) domain, the intermediate catalytic core domain, and UBL1, 2, 3, 4, and 5 (C-terminal ubiquitin-like domains) [17,18] (Figure 1). Through its deubiquitinase activity, HAUSP has been reported to control the activity of several oncogenic transcription factors, including NOTCH1 in T-cell acute lymphoblastic leukemia [19,20], N-Myc in neuroblastoma [16], β-catenin in colorectal cancer [21], and NEK2 in multiple myeloma [22], indicating that HAUSP has an oncogenic role in cancer. In several tumors, HAUSP was shown to bind to and deubiquitinate the E3 ubiquitin ligases, MDM2 [23,24,25] and MDMX [26], which are known negative regulators of the tumor suppressor gene *p53* [27,28,29,30]. Blocking the deubiquitination function of HAUSP enabled the ubiquitination and proteasomal degradation of both MDM2 and MDMX, which subsequently stabilized and restored p53 protein levels to induce cell death [23,24,26].

HAUSP, via its different domains, interacts with several proteins implicated in coordinating various signaling pathways. HAUSP is found in many protein complexes with different functions, including the “Epigenetic Code Replication Machinery,” ECREM. Indeed, HAUSP is a main partner of ECREM, a large macromolecular complex that includes several epigenetic players, such as the ubiquitin-like containing plant homeodomain (PHD) and an interesting new gene (RING) finger domains 1 (UHRF1), as well as DNA methyltransferase 1 (DNMT1), histone deacetylase 1 (HDAC1), histone methyltransferase G9a, and histone acetyltransferase TIP60 [32,33,34,35,36,37,38]. Growing evidence indicates that the silencing of tumor suppressor genes (TSGs) in tumors is the result of a coordinated in-depth dialogue between DNA methylation and various histone post-translational modifications (PTMs) [39,40,41,42].

Several reports have shown that a faithful inheritance of the epigenetic patterns (DNA methylation and histone PTMs) during cell division involves temporal and spatial control of the chromatin effector proteins mainly UHRF1 and DNMT1, which govern various epigenetic events [34,39,43,44,45,46,47]. Understanding the major factors regulating the expression and activity of chromatin effector proteins UHRF1 and DNMT1 could therefore unlock new secrets regarding the transmission of epigenetic patterns during cell division. HAUSP positions itself at the top of the regulatory hierarchies involved in the epigenetic silencing of TSGs in cancer due to its deubiquitinase activity and ability to team up through direct interactions with several epigenetic players (Table 1) mainly the epigenetic reader UHRF1 [38,48]; DNMT1 [38,49]; the histone acetyltransferases TIP60 [50] and CBP (CREB binding protein) [51]; the histone lysine methyltransferases EZH2 (Enhancer of Zeste 2) [52] and MLL5 (Mixed-lineage leukemia 5) [53], and the lysine specific histone demethylases LSD1 (Lysine-specific demethylase 1) [54,55] and PHF8 (PHD finger protein 8) [56] (Figure 2). The present review highlights the increasingly evident role of HAUSP as an epigenetic master regulator that governs a set of epigenetic players involved in the maintenance of DNA methylation and histone PTMs.

## 2. Role of HAUSP in the Maintenance of DNA Methylation

Mammalian DNA methylation patterns are established and faithfully copied to daughter cells during replication by the maintenance DNMT1, whereas DNA methylation during embryonic development is catalyzed by the de novo expression of DNA methyltransferases DNMT3a and DNMT3b [58,59]. DNMT1 is overexpressed in many types of cancer and plays an important role in tumorigenesis through the epigenetic silencing of many TSGs [60,61,62]. The expression of DNMT1 is regulated by post-translational modifications, including ubiquitination and acetylation [36,38,49,63,64].

### 2.1. HAUSP-Dependent Regulation of DNMT1 by Ubiquitination

The deubiquitinase activity of HAUSP regulates the stability and enzymatic activity of DNMT1 through several mechanisms. HAUSP can bind to DNMT1 and stabilize it through HAUSP-mediated deubiquitination [63]. High expression levels of DNMT1 protein have been found in human colon cancers, and this overexpression was correlated with HAUSP protein expression. Interestingly, the depletion of HAUSP in both human embryonic kidney cells and colorectal cancer cells resulted in an increase in DNMT1 ubiquitination and a reduction in DNMT1 protein expression. A similar increase in DNMT1 ubiquitination levels and a decrease in its protein expression were reported when HAUSP was knocked out in colorectal cancer cells [63]. By contrast, HAUSP overexpression in HAUSP knockout cells led to deubiquitination of DNMT1 and restoration of DNMT1 protein expression, suggesting that the deubiquitinase activity of HAUSP protects DNMT1 from proteasomal degradation, thereby promoting the stability and enzymatic activity of DNMT1 and the maintenance of DNA methylation [63].

One of the well-documented epigenetic regulators of the DNA methylation maintenance machinery is the epigenetic reader UHRF1, which has multiple functional domains [65,66,67]. Through its SRA domain, UHRF1 recognizes and binds hemi-methylated CpG islands, and via the same SRA domain, UHRF1 recruits DNMT1 to its correct position on chromatin to ensure faithful transmission of DNA methylation patterns during DNA replication [68,69,70]. Besides its interaction with DNMT1, UHRF1 also interacts directly with other epigenetic writers, such as G9a [71] and TIP60 [36,72,73], and epigenetic erasers, including HDAC1) [74] and HAUSP [36,38,75] to form a macromolecular complex. In this complex, the deubiquitinase HAUSP can function as a hub protein through its direct interaction with UHRF1, which leads to its deubiquitination protecting UHRF1 from proteasomal degradation, thereby allowing UHRF1 to read epigenetic marks and to recruit and guide the right enzyme, writer, or eraser to the correct position at the right time to catalyze the target epigenetic mark [39,45,75,76,77]. In this way, UHRF1 ensures a faithful inheritance of both types of epigenetic marks (DNA methylation and histone PTMs) to daughter cells and leads to the epigenetic silencing of several TSGs, such as *p16^INK4A^*, *PTEN*, *BRCA1*, and *hMLH1* [32,78].

Several studies have reported that HAUSP, via its deubiquitinase activity, can regulate the stability and expression of UHRF1, as well as DNMT1, and protect both proteins from ubiquitination-mediated proteasomal degradation, thereby maintaining DNA methylation [36,38,57,75]. HAUSP has been shown to interact directly with DNMT1 and UHRF1 to form a trimeric complex on chromatin [38]. In this complex, HAUSP, via its C-terminal domain, interacted with the targeting sequence (TS) domain of DNMT1, whereas HAUSP, through its TRAF domain, interacted with the SRA domain of UHRF1, thereby promoting the stability of UHRF1 and DNMT1 through HAUSP-dependent deubiquitination (Figure 3) [38]. Moreover, the HAUSP/DNMT1/UHRF1 complex was located at promoter regions of several TSGs, including *SFRP1*, *IGFBP3*, *HHIP*, and *HOXA7* [38], which are known to undergo epigenetic silencing in various tumors [79,80]. Interestingly, a depletion of HAUSP was accompanied by decreased expression levels of both UHRF1 and DNMT1 proteins and a subsequent marked demethylation of target genes at their promoters [38]. These data suggest that HAUSP plays a key role in the regulation of DNMT1-dependent DNA methylation. HAUSP first interacts with DNMT1 and forms a dimeric complex. UHRF1 then joins the HAUSP/DNMT1 dimer and recruits the dimeric complex to the sites of methylation by forming a trimeric complex on the promoters of the silenced genes. In the trimeric complex, HAUSP plays a dual role by directly stimulating the enzymatic activity of DNMT1 and by regulating the protein stability of UHRF1.

HAUSP binds to UHRF1 by its TRAF domain, but it also binds via its UBL domain (residues 560–664) to a linker region encompassing amino acids 600–687 between the SRA and RING finger domains of UHRF1, and this direct interaction is required for UHRF1 stability [48]. The downregulation of HAUSP decreased the expression levels of UHRF1 protein, whereas the overexpression of wild-type HAUSP, but not the catalytically inactive mutant HAUSP (C223S), significantly decreased the ubiquitination of UHRF1 and promoted its stability [48]. These findings implicate HAUSP, via its deubiquitinase activity, in the deubiquitination and stability of UHRF1 and DNMT1 and suggest that the epigenetic silencing of TSGs detected in many tumors is due to HAUSP-mediated stabilization of the UHRF1 and DNMT1 proteins (Figure 3).

Obtaining a deep insight into how HAUSP deubiquitinase activity is involved in the stabilization of UHRF1 protein will help in understanding the regulatory role of HAUSP in the UHRF1-dependent maintenance of DNA methylation. In this regard, HAUSP was shown to regulate the stability of UHRF1 protein by targeting the ubiquitin ligase activity of the UHRF1 RING domain [38,65,81], which is used by UHRF1 to ubiquitinate itself (autoubiquitination) [65,66] or to ubiquitinate other substrates, mainly histone 3 [81,82,83]. Indeed, HAUSP, via its deubiquitinase activity, was shown to interfere with the E3 ubiquitin ligase activity of the RING domain of UHRF1, thereby eliminating the autoubiquitination of UHRF1 via the removal of ubiquitin adducts [38]. This led to the stabilization of UHRF1, indicating that HAUSP regulates the maintenance of DNA methylation through a direct association with UHRF1 [38].

### 2.2. HAUSP-Dependent Regulation of DNMT1 by Acetylation

HAUSP can also regulate DNMT1 stability through an acetylation process [36,38,49,63,64]. HAUSP, through its UBL domains (residues 560–1102), was found to stabilize DNMT1 by binding to the KG linker (KG^DNMT1^) of DNMT1 (residues 1109–1119) promoting the stability of DNMT1 [49]. The depletion of HAUSP in several cancer cell lines resulted in a significant decrease in DNMT1 expression at the protein level, but not at the mRNA level. Similar findings were observed when HAUSP was knocked out in colon cancer cells, supporting the concept that HAUSP deubiquitinates and protects DNMT1 from proteasomal degradation, thereby promoting DNMT1 stability [49]. Interestingly, the inhibition of histone deacetylase 1 (HDAC1), or the overexpression of histone acetyltransferase TIP60, increased the acetylation of the KG linker of DNMT1 and impaired DNMT1 binding to HAUSP [49]. These findings indicate that the KG linker of DNMT1 is indispensable for DNMT1 interaction with HAUSP. Hence, the acetylation of the KG linker of DNMT1 impairs the interaction between DNMT1 and HAUSP, thereby stimulating the degradation of the DNMT1 protein [49].

## 3. Role of HAUSP in the Regulation of Histone Post-Translational Modifications

Abnormal histone PTMs, such as methylation, acetylation, and ubiquitination, can drive the epigenetic silencing of TSGs in cancers [84,85,86,87]. Histone PTMs are decoded, deposited, and erased by a specific set of enzymes. A better understanding of the regulatory mechanisms of these enzymes is essential to fully decipher the complexity of the epigenetic determinants of TSG silencing in cancer. The deubiquitinase activity of HUASP can regulate the key histone readers and modifiers that connect histone PTMs with DNA methylation.

### 3.1. Role of HAUSP in the Regulation of Histone Monoubiquitination

Monoubiquitination of histones, particularly H2A and H2B, is a common type of histone PTM that controls gene expression [84,88,89]. Monoubiquitination of histone H2A on lysine 119 (H2AK119ub1) is a well-documented epigenetic modification linked to the Polycomb Repressor Complex 1 (PRC1) and is frequently associated with gene silencing [89,90,91,92]. This histone mark (H2AK119ub1) is catalyzed by the E3 ubiquitin ligase RING1B, which is overexpressed in many human tumors [93,94,95,96] and plays an important role in PRC1-mediated gene silencing through monoubiquitination of histone H2A [97,98]. HAUSP can directly interact with and specifically deubiquitinate RING1B, thereby modulating the level of ubiquitinated H2A [97]. The same effects on H2AK119ub1 were found when HAUSP was downregulated or knocked out in human colorectal carcinoma cells HCT116 [99]. Indeed, both depletion and the knockout of HAUSP in HCT116 cells resulted in a decrease in the expression of RING1B and ubiquitination levels of H2AK119. This effect was associated with an increase in the expression of the tumor suppressor gene *p16^INK4A^* and cell proliferation inhibition, indicating that HAUSP has a key regulatory role in the monoubiquitination of H2A that is associated with gene inhibition [99]. Besides its regulatory effects on the monoubiquitination of histone H2, HAUSP has been reported to induce a selective deubiquitination of H2B and this mechanism requires the binding of HAUSP to guanosine 5′-monophosphate synthetase (GMPS) [100].

### 3.2. Role of HAUSP in the Regulation of Histone Acetylation

The histone acetyltransferase TIP60 is a member of the ECREM complex that also includes HAUSP, UHRF1, and DNMT1, and each member has regulatory effects on the other partners (35, 49, 61, 70, 71, 73). TIP60, through its acetyltransferase, was shown to acetylate, destabilize, and trigger the ubiquitination of DNMT1 through the E3 ligase activity of UHRF1 [63]. TIP60 is also required for acetylation of histones [101,102,103] and of nonhistone proteins, such as the tumor suppressor p53 [104,105]. Thus, deciphering the mechanisms of TIP60 regulation is crucial for unraveling the complex role of this histone acetyltransferase in tumors. TIP60 is a target of the deubiquitinase activity of HAUSP [50], which interacts through its TRAF domain to deubiquitinate and stabilize TIP60, leading to the subsequent induction of p53-dependent apoptosis [50]. Interestingly, the overexpression of wild-type HAUSP deubiquitinated and stabilized TIP60, whereas mutation of the catalytic cysteine residue (C223S) of HAUSP abolished the deubiquitination and promoted the degradation of TIP60 [50]. By contrast, depletion of HAUSP decreased the expression levels of the TIP60 protein, resulting in a decrease in the acetylation of key TIP60 substrates, including histones H2A at Lys 5 (H2AK5) and H4 at Lys 5 (H4K5), as well as p53 [50]. The use of P22077, a specific HAUSP inhibitor [106], also led to the destabilization and degradation of TIP60 and subsequent inhibition of apoptosis [50]. These findings indicate that HAUSP deubiquitinase activity is a prerequisite for the stabilization and acetyltransferase activity of Tip60 and enables the acetylation of the key TIP60 substrates H2AK5, H4AK5, and p53.

HAUSP can also regulate the acetylation of histone H3 at Lys 56 (H3K56Ac) by targeting the CBP histone acetyltransferase [51] which is overexpressed in cancer [107,108,109]. HAUSP, through a region encompassing amino acids 600–687, was found to interact with the CH3 domain of CBP to enhance the CPB histone acetyltransferase activity [51]. Conversely, downregulation of HAUSP reduced the expression levels of H3K56Ac in breast cancer cells and in head and neck cancer cells, indicating that the CBP histone acetyltransferase requires the deubiquitinase activity of HAUSP for the specific acetylation of histone H3 at Lys 56 [51].

### 3.3. Role of HAUSP in the Regulation of Histone Methylation

Histone lysine methyltransferases (KMTs), such as EZH2 [110] and MLL5, [53] and lysine specific histone demethylase (KDMs), including LSD1 [111,112] and PHF8 [113,114], are two families of enzymes involved in histone methylation and demethylation, respectively. Specifically, the trimethylation of lysine 27 of histone 3 (H3K27me3), mediated by EZH2, is considered one of histone changes that silence target genes, thereby promoting tumor growth and metastasis in many tumors [110,115,116,117]. EZH2 is a core subunit of the Polycomb Repressive Complex 2 (PRC2), which is associated with cancer [118,119].

Recently, EZH2 was identified as a novel HAUSP partner in human U2OS osteosarcoma cells and human HCT116 colorectal cancer cells [52]. The HAUSP–EZH2 interaction is mediated by the C-terminal region of HAUSP and the ^489^PRKKKRK^495^ region in EZH2 [52]. Both the depletion of HAUSP and its knockout in HCT116 cells reduced the expression levels of EZH2 protein and H3K27me3, whereas HAUSP reintroduction to HAUSP knockout HCT116 cells increased the protein expression levels of EZH2 [52]. These findings indicate that HAUSP positively regulates the protein expression and activity of EZH2 and that the binding of EZH2 to HAUSP is essential both for the stability of EZH2 and for its function as an epigenetic writer of the trimethylation of H3 at K27 in cancer cells.

The trimethylation of histone 3 at Lys 4 (H3K4m3) is specifically mediated by histone 3 methyltransferase MLL5 [120,121]. High levels of H3K4m3 have been detected in several tumors, including renal cell carcinoma [122] and hepatocellular carcinoma [123], and this high expression was correlated with shorter survival of patients and a higher chance of tumor recurrence [124]. HAUSP via its deubiquitinase activity was found to regulate the expression levels of H3K4m3 by targeting MLL5 [53]. HAUSP through its TRAF domain was found to bind to and deubiquitinate MLL5 leading to the stability of MLL5 [53]. Depletion of HAUSP was shown to promote the degradation of MLL5 protein, whereas HAUSP overexpression decreased the ubiquitination of MLL5 and increased its stability. By contrast, a catalytically inactive C223S mutant HAUSP had no effect on MLL5 ubiquitination or stability, suggesting that the deubiquitinase activity of HAUSP is important for both deubiquitination and stabilization of the MLL5 protein [53].

Similarly, the methylation of histones is constantly opposed by demethylases, which act to remove the methyl moieties from histone substrates. One of the well-documented demethylases is LSD1 (also known as KDM1A), which plays a key regulatory role in gene expression by removing the monomethyl and dimethyl groups from the methylated lysine 4 of histone H3 and the methylated lysine 9 of histone H3 [111,112,125]. Several studies have reported that LSD1 functions as an oncogene through regulating several pathways [126,127,128]. Thus, understanding how LSD1 is upregulated in cancer is important for understanding the biological activity of this enzyme. In this context, LSD1 was found to act as a specific substrate of the deubiquitinase activity of HAUSP, and the interaction between the two enzymes is involved in glioma progression [54]. High expression levels of both LSD1 and HAUSP were detected in brain tissues from 150 patients with glioma compared to normal brain tissues, and the elevated expression was correlated with glioma progression [54]. By contrast, depletion of HAUSP decreased the protein levels of LSD1 without changing the LSD1 mRNA levels, and this effect was associated with an inhibition of glioblastoma cell invasiveness [54]. Interestingly, the overexpression of wild-type HAUSP, but not the catalytically inactive mutant HAUSP (C223A), increased the protein levels of LSD1 in the A172 and T98G glioblastoma cell lines, [54] suggesting that the expression and deubiquitinase activity of HAUSP is a main cause of the high expression levels of LSD1 found in tumors. In agreement with this hypothesis, LSD1 was overexpressed in samples of human breast tumors and regulated by the deubiquitinase activity of HAUSP [55]. HAUSP bound to a region encompassing amino acids 520–852 of the C-terminal of LSD1, leading to the deubiquitination of LSD1 and the promotion of breast cancer tumorigenesis and metastasis [55]. The depletion of HAUSP in breast cancer cells or the use of P5091, a HAUSP inhibitor, resulted in a dramatic increase in LSD1 ubiquitination, a decrease in LSD1 protein levels, and an increase in the methylation of H3K4me2 and H3K9me2. These findings demonstrate that HAUSP deubiquitinates and stabilizes LSD1, thereby promoting the demethylase activity of LSD1 and increasing the demethylation of H3K4me2 and H3K9me2 [55].

The demethylase PHF8 (also known as KDM7B) catalyzes the demethylation of both mono- or dimethylated H3K9 and H3K27 [113] and binds histone H3 lysine 4 tri-methyl (H3K4me3) [114,129]. Like the LSD1 demethylase, PHF8 is overexpressed in many types of tumors and plays an oncogenic role [129,130,131]. Thus, discovering the upstream events involved in the regulation of PHF8 is important for understanding how this enzyme is stabilized and upregulated in cancer. PHF8 was found to be regulated and stabilized through the deubiquitinase activity of HAUSP [56]. HAUSP via its TRAF-like domain interacted with the C-terminal region of PHF8 promoting deubiquitination and stabilization of PHF8, which resulted in an upregulation of cyclin A2, a key cell cycle regulator in cancer [56]. Depletion of HAUSP decreased the expression of the PHF8 protein. Moreover, depletion of either HAUSP or PHF8 was associated with increased levels of H3K9me1, H3K9me2, H3K27me2, and H4K20me1, indicating that HAUSP controls the stability and demethylase activity of PHF8 and hence the methylation of H3 and H4 [56].

### 3.4. Role of HAUSP in the UHRF1-Mediated Readout of Histone H3 Lysine 9 Methylation

During DNA replication in the S phase of the cell cycle, UHRF1 is loaded onto replication forks via its binding affinity to hemi-methylated DNA through its SRA domain [68,69,70] and by its ability to bind to and ubiquitinate di-or tri-methylated histone H3K9 (H3K9me2,3) through a coordinated cooperation between both its TTD and PHD domains [77,132]. However, in-depth investigation of how UHRF1-mediated ubiquitination of methylated histone H3K9 is regulated is of great importance for understanding the regulation of this maintenance machinery axis.

A structural characterization of the HAUSP/UHRF1 complex showed that HAUSP, through its first two UBL domains (UBL1-2), binds to a polybasic region (PBR) in the C terminus of UHRF1 [57], a region responsible for blocking the interaction of the TTD domain of UHRF1 with the tri-methylation of lysine 9 of histone 3 (H3K9me3) [133]. The binding of HAUSP to the PBR region of UHRF1 induces an allosteric regulation of UHRF1 by disrupting the intramolecular interaction between the TTD and PBR domains of UHRF1, thereby releasing the TTD domain from its association with PBR and thus from the “TTD-occluded state” and allowing it to adopt an “open state” that promotes the association of the TTD-PHD domains of UHRF1 with chromatin and, hence, efficient H3K9me3 binding [57]. Interestingly, the dissociation of HAUSP from UHRF1 through the introduction of a HAUSP-interaction-defective mutation in UHRF1 reduced the UHRF1 association with chromatin [57]. These findings indicate that HAUSP, through its deubiquitinase activity and its interaction with UHRF1, plays a main role in the regulation of H3K9me3 via a dual regulatory role, first by inducing UHRF1 deubiquitination and then by promoting the association of UHRF1 with chromatin.

## 4. HAUSP Inhibitors as Promising Anticancer Agents

The important role of HAUSP in modulating several pro-proliferative and anti-apoptotic genes has led to extensive efforts to develop HAUSP-selective inhibitors [134,135,136]. Moreover, the solved X-ray structure of HAUSP in complex with those molecules further advanced the structure-based design potent and reversible and irreversible inhibitors [137]. Irreversible HAUSP-small molecules inhibitors work via alkylating the catalytic Cys223, blocking its interaction with ubiquitin. However, the similarity of the catalytic site of HAUSP with other deubiqutinases reduces the selectivity of such inhibitors. On the other hand, reversible inhibitors that bind noncovalently to the site adjacent to the catalytic domain displayed better selectivity vis-à-vis other deubiquitinases [106]. HAUSP inhibitors and their anticancer activities have been recently reviewed [137].

## 5. Conclusions

HAUSP has been shown to deubiquitinate and stabilize several pro-proliferative and anti-apoptotic genes in many human malignancies, causing apoptosis inhibition and enhanced cell proliferation. Via its structural domains, HAUSP interacts with the epigenetic reader UHRF1, several writers such as DNMT1, TIP60, CBP, EZH2, and MLL5, and with erasers, including LSD1 and PHF8 (Figure 2). Under HAUSP deubiquitinase activity, members of this specific set of enzymes work together to catalyze epigenetic marks (DNA methylation and histone PTMs), ensuring a faithful inheritance of both types of epigenetic marks to daughter cells and leading to the epigenetic silencing of several TSGs (Figure 4). This review highlighted the increasing role of HAUSP as a key epigenetic regulator of the chromatin effector proteins and thus the importance of inhibiting HAUSP-dependent deubiquitination as a promising antitumor strategy. Therefore, understanding the mechanisms involved in the regulation of HAUSP function will allow to find new targets and explore new drugs to inhibit the expression and or activity of HAUSP, allowing cancer cells to undergo apoptosis through a reactivation of several TSGs genes.

## Figures and Tables

**Figure 1 genes-13-00042-f001:**
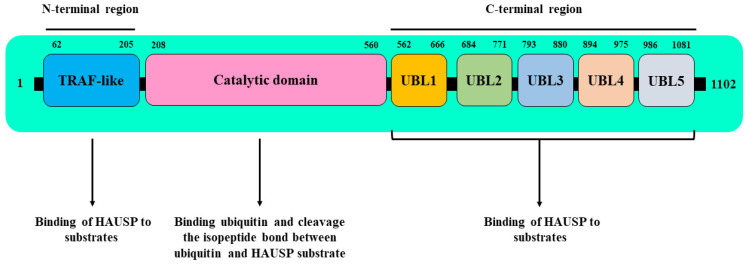
Schematic representation of the domain structure of HAUSP (herpes virus-associated ubiquitin-specific protease). The structure of HAUSP includes seven domains, N-terminal TRAF-like (Tumor necrosis factor Receptor–Associated Factor) domain, intermediate catalytic core and five consecutive C-terminal ubiquitin-like domains (Ubiquitin-like domains), UBL1-5. The N-terminal TRAF-like domain and the five UBL domains are sites for the binding of HAUSP to many proteins. Through the catalytic core domain, HAUSP binds ubiquitin and cleave the is peptide bond between ubiquitin and HAUSP substrate [17,18,31].

**Figure 2 genes-13-00042-f002:**
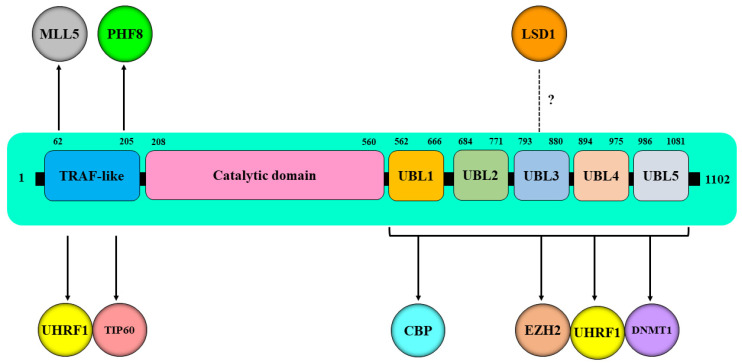
Schematic representation of interactions of HAUSP domains with various epigenetic players. HAUSP via its TRAF-like domain interacts with the ubiquitin-like containing plant homeodomain (PHD) and an interesting new gene (RING) finger domains 1 (UHRF1) [38], histone acetyltransferase TIP60 [50], the histone lysine methyltransferase MLL5 (Mixed-lineage leukemia 5) [53] and the lysine specific histone demethylase PHF8 (PHD finger protein 8) [56]. HAUSP through the C-terminal region which covers its five UBL domains can interact also with UHRF1 [48], histone acetyltransferase CBP (CREB binding protein) [51], histone methyltransferase EZH2 [52] and DNA methyltransferase 1 (DNMT1) [38]. HAUSP can also bind to LSD1 but the interaction site is not yet known [54].

**Figure 3 genes-13-00042-f003:**
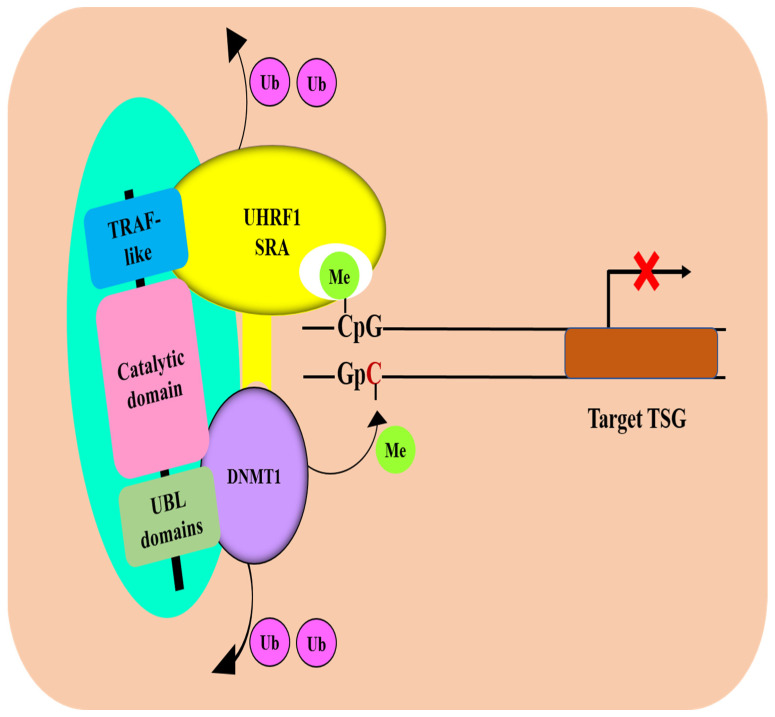
Schematic representation of role of HAUSP in DNA methylation maintenance through regulating the stability UHRF1/DNMT1 dual. HAUSP, via its UBL domains, interacts with the targeting sequence (TS) domain of DNMT1, whereas HAUSP, through its TRAF-like domain, interacted with the SET- and RING-associated (SRA) domain of UHRF1, thereby promoting the stability of UHRF1 and DNMT1 through HAUSP-dependent deubiquitination [38]. The trimeric complex located at promoter regions of target TSGs on chromatin. During DNA replication, the SRA domain of UHRF1 can read methylated CpG sites (hemi-methylated DNA) located within the promoter regions and via the same SRA domain, UHRF1 recruits DNMT1 and guide it to methylate the “unmethylated” cytosine of the newly synthetized DNA strand leading to hypermethylation of TSG promoter.

**Figure 4 genes-13-00042-f004:**
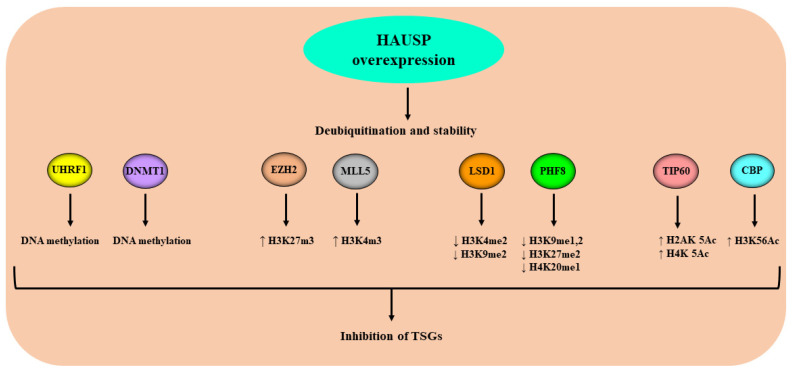
Schematic representation of the effect of HAUSP overexpression in cancer on the chromatin effector proteins and the related downstream events. High expression of HAUSP in cancer deubiquitinate and stabilizes the chromatin effector proteins causing abnormal DNA methylation and histone post-translational modifications which contribute to the silencing epigenetic of TSGs.

**Table 1 genes-13-00042-t001:** Role of HAUSP in the regulation of epigenetic players and related events.

Epi-Partner	Role of Epi-Partner	HAUSP Interaction Site	Epi-Partner Interaction Site	Related Epigenetic Events	Refs.
UHRF1	Reader of both epigenetic marks (DNA methylation and histone code)	UBL1 domain (residues 560–664)	A linker region encompassing amino acids 600–687 between the SRA and RING finger domains of UHRF1	Promoting the stability of UHRF1 through HAUSP-dependent deubiquitination	[48]
		TRAF-like domain	SRA domain	Promoting the stability of UHRF1 through HAUSP-dependent deubiquitination	[38]
		(UBL1-2) domains	A polybasic region (PBR) in the C terminus	Promoting the association of the TTD-PHD domains of UHRF1 with chromatin and, hence, efficient H3K9me3 binding	[57]
DNMT1	DNA methyltransferase 1	C-terminal domain	Targeting sequence (TS) domain	Promoting the stability of DNMT1 through HAUSP-dependent deubiquitination	[38]
		UBL domains (residues 560–1102)	KG linker (residues 1109–1119)	Promoting the stability of DNMT1 through acetylation of KG linker of DNMT1	[49]
TIP60	Histone acetyltransferase	TRAF-like domain		Increased levels of H2AK 5 and H4K5	[50]
CBP	Histone acetyltransferase	Region encompassing amino acids 600–687	CH3 domain	Increased levels of H3K56Ac	[51]
MLL5	Histone lysine methyltransferase	TRAF domain	Multiple domains	Increased levels of H3K4m3	[53]
EZH2	Histone lysine N-methyltransferase	C-terminal region	^489^PRKKKRK^495^ region	Increased levels of H3K27m3	[52]
LSD1	Lysine specific demethylase 1				[54]
			A region encompassing amino acids 600–687	Demethylation of H3K4me2 and H3K9me2	[55]
PHF8	Histone lysine demethylase	TRAF-like domain	The C-terminal region	Demethylation of H3K9me1,2, H3K27me2, and H4K20me1	[56]

## Data Availability

Not applicable.

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
