# Peer review of "HAUSP Is a Key Epigenetic Regulator of the Chromatin Effector Proteins"

_genes, 2021, doi:10.3390/genes13010042_

Round 1
Reviewer 1 Report
A specific comments on very minor issues/preferences:
1) UHRF1 is described as a "new gene", it's not entirely what is meant by this, so some clarification would be beneficially. It's certainly not newly discovered.
2) Should be made clear that DNMT1 protein levels are being referred to, not mRNA, when using the term expression in the experiments described in 101-113). (Though this is obviously the case given the context of the paragraph). This is also the case forr some of the other proteins refered to in the text eg. EZH2.
3) in 122 It's not the deubiquitination activities here that would cause HAUSP to be described as a "hub", more it's ability to interact with multiple chromatin modifying enzymes in the same complex and the facts that bringing together of chromatin modifying enzymes somehow enhances their functionality, beyond protection from proteosomal degradation, which conceivably does not require more than a 2 way interaction with HAUSP and the chromatin modifier. You refer to an explanation as to why a trimeric complex is required for the maintenance of DNA methylation in the figure (relating to binding on each strand), so I mainly have an issue with the ordering of the text relating to this point.
4) in 165, worth mentioning deubiquitination could conceivably be achieved in dimers with each protein, so it's not a satisfactory explanation as to why the complex exists in a trimer and hints to additional, non-catalytic functions of HAUSP. Which indeed you cover when referring to the role of HAUSP and UHRF1's association with chromatin later in the text.
5) I thought the review was clearly written, however is quite formulaic and repetitious in it's presentation of evidence for HAUSP's role in the stability of the particular proteins covered. This is a missed opportunity to highlight the common theme, .i.e regulation of the stability of many important epigenetic modifiers via deubiquitilition, that all seemed to be overexpressed in a cancer context, and refer more to the TSGs regulated by the chromatin modifiers and the extent to which exclusively TSGs are suppressed and maybe even the features that TSGs possess that make them susceptible to epigenetic silencing by proteins regulated by HAUSP.
Author Response
Answers to reviewer 1:
The authors would like to thank reviewer 2 for his (her) helpful comments and suggestions to improve the present article.
Reviewer #1:
specific comments on very minor issues/preferences:
Major Points
Comment 1. UHRF1 is described as a "new gene", it's not entirely what is meant by this, so some clarification would be beneficially. It's certainly not newly discovered.
Authors response:
We agree with the reviewer that UHRF1 is not a new gene. The term "new gene" was used in the text for the RING domain of UHRF1 not for UHRF1 itself. RING domain is referred to the interesting new gene as indicated in the text.
Comment 2. Should be made clear that DNMT1 protein levels are being referred to, not mRNA, when using the term expression in the experiments described in 101-113). (Though this is obviously the case given the context of the paragraph). This is also the case for some of the other proteins referred to in the text eg. EZH2.
Authors response: As requested by the reviewer and to avoid any confusion for readers, we added protein expression for DNMT1 in the paragraph (101-113) and for EZH2 in the paragraph (258-265).
Comment 3. in 122 It's not the deubiquitination activities here that would cause HAUSP to be described as a "hub", more it's ability to interact with multiple chromatin modifying enzymes in the same complex and the facts that bringing together of chromatin modifying enzymes somehow enhances their functionality, beyond protection from proteasomal degradation, which conceivably does not require more than a 2 way interaction with HAUSP and the chromatin modifier. You refer to an explanation as to why a trimeric complex is required for the maintenance of DNA methylation in the figure (relating to binding on each strand), so I mainly have an issue with the ordering of the text relating to this point.
Authors response:
This information has been edited in page 5 line 122 to highlight the role of HUASP described as a hub protein by ensuring direct interaction with UHRF1 in the EXREM complex.
The paragraph (114-119) describes the well-documented role of the epigenetic reader UHRF1 through its direct interactions with multiple chromatin modifying enzymes in the DNA methylation maintenance machinery and how HAUSP via its direct interaction with UHRF1 and its ability to stabilize UHRF1 protein can be involved in this machinery. In the next paragraph (130-148) we illustrated how HAUSP forms a trimeric complex with UHRF1 and DNMT1 and how HAUPS through these interactions and its deubiquitinase activity can play an important role in DNA methylation maintenance through regulating the stability UHRF1/DNMT1 dual (Figure 3).
Comment 4. in 165, worth mentioning deubiquitination could conceivably be achieved in dimers with each protein, so it's not a satisfactory explanation as to why the complex exists in a trimer and hints to additional, non-catalytic functions of HAUSP. Which indeed you cover when referring to the role of HAUSP and UHRF1's association with chromatin later in the text.
Authors response:
Felle et al (Nucleic Acids Res. 2011) have shown that HAUSP is able to interact with both UHRF1 and DNMT1 protecting them from proteasomal degradation. HAUSP via its UBL domains, interacts with the targeting sequence (TS) domain of DNMT1, whereas HAUSP, through its TRAF-like domain, interacted with the SET- and RING-associated (SRA) domain of UHRF1, thereby promoting the stability of UHRF1 and DNMT1 through HAUSP-dependent deubiquitination (Figure 3). Felle et al have also shown that the trimeric complex located at promoter regions of target TSGs on chromatin. They have identified HAUSP as the first factor that has a stimulatory effect on the enzymatic activity of DNMT1 and as a regulator of UHRF1 protein stability, hence directly affecting UHRF1-dependent DNA methylation efficiency. Based on these findings and other studies (Sharif et al, Nature 2007, Bostick et al, Science 2007 Achour et al, Oncogene 2008), we suggested a schematic representation of role of HAUSP in DNA methylation maintenance through regulating the stability UHRF1/DNMT1 dual (Figure 3). Figure 3 shows that HAUSP and DNMT1 can form a soluble dimer complex that associate with UHRF1 as a trimeric complex on chromatin. During DNA replication, the SRA domain of UHRF1 can read methylated CpG sites (hemi-methylated DNA) located within the promoter regions and via the same SRA domain, UHRF1 recruits DNMT1 and guide it to methylate the “unmethylated “cytosine of the newly synthetized DNA strand leading to hypermethylation of TSG promoter. To clarify this point regarding the trimeric complex and to avoid any misleading for readers, we edited Figure 3 to show the interactions between the three proteins HAUSP, UHRF1 and DNMT1 and the domains involved. In the edited figure 3, the catalytic domain of HAUSP covers both UHRF1 and DNMT1.
Comment 5. I thought the review was clearly written, however is quite formulaic and repetitious in it's presentation of evidence for HAUSP's role in the stability of the particular proteins covered. This is a missed opportunity to highlight the common theme, .i.e regulation of the stability of many important epigenetic modifiers via deubiquitilition, that all seemed to be overexpressed in a cancer context, and refer more to the TSGs regulated by the chromatin modifiers and the extent to which exclusively TSGs are suppressed and maybe even the features that TSGs possess that make them susceptible to epigenetic silencing by proteins regulated by HAUSP.
Authors response:
HAUSP is found in many protein complexes with different functions, including the “Epigenetic Code Replication Machinery,” ECREM. Our review summarized in particularly the increasing role of HAUSP as a key epigenetic regulator of the chromatin effector proteins mainly the epigenetic reader UHRF1, DNA methyltransferase 1 (DNMT1), histone acetyltransferases TIP60 and CBP, histone methyltransferases EZH2 and MLL5 and lysine specific histone demethylases LSD1 and PHF8 (Figure 2). In the review, we also discussed the related downstream events especially the epigenetic silencing of tumor suppressor genes in cancer (Figure 3) and thus the importance of inhibiting HAUSP-dependent deubiquitination as a promising antitumor strategy.
Reviewer 2 Report
Manuscript entitled “HAUSP is a key epigenetic regulator of the chromatin effector 2 proteins“ is dealing with very interesting topic and revealing mechanisms of herpes virus-associated ubiquitin-specific protease in the maintainance of DNA methylation via acetylation proceses, the regulation of histone monoubiquitination, acetylation and regulation of histon methylation. It is envolved in deubiquitation and stabilization of pro apoptotic genes.
Review is well structured, detailed and nicely prepared.
Small molecule inhibitors targeting the deubiquitinase activity of HAUSP have been developed so it would add to the value of this paper if separated paragraph abouth the therapy topic would be mentioned.
Author Response
Answers to reviewer 2:
The authors would like to thank reviewer 2 for his (her) helpful comments and suggestions to improve the present article.
Reviewer #2:
Manuscript entitled “HAUSP is a key epigenetic regulator of the chromatin effector proteins“ is dealing with very interesting topic and revealing mechanisms of herpes virus-associated ubiquitin-specific protease in the maintenance of DNA methylation via acetylation process, the regulation of histone monoubiquitination, acetylation and regulation of histone methylation. It is involved in deubiquitation and stabilization of pro apoptotic genes. Review is well structured, detailed and nicely prepared.
Comment 1. Small molecule inhibitors targeting the deubiquitinase activity of HAUSP have been developed so it would add to the value of this paper if separated paragraph about the therapy topic would be mentioned.
Authors response: As requested by the reviewer, we have included a new paragraph in the revised version to highlight inhibiting HAUSP using small molecule inhibitors targeting its deubiquitinase activity as a promising antitumor strategy.